# Modelling disease mitigation at mass gatherings: A case study of COVID-19 at the 2022 FIFA World Cup

Martin Grunnill[1,¤]*, Julien Arino[2], Zachary McCarthy[1], Nicola Luigi Bragazzi[1], Laurent Coudeville[3], Edward W. Thommes[3,4], Amine Amiche[5], Abbas Ghasemi[6,7], Lydia Bourouiba[6], Mohammadali Tofighi[8,9], Ali Asgary[9,10], Mortaza Baky-Haskuee[9], Jianhong Wu[1,10¤]*

1 Laboratory of Industrial and Applied Mathematics (LIAM), York University, Toronto, Ontario, Canada, 2 Department of Mathematics, University of Manitoba, Winnipeg, Manitoba, Canada, 3 Modeling, Epidemiology and Data Science (MEDS), Sanofi, Lyon, France, 4 Department of Mathematics and Statistics, University of Guelph, Guelph, Ontario, Canada, 5 Sanofi, Dubai, United Arab Emirates, 6 The Fluid Dynamics of Disease Transmission Laboratory, Massachusetts Institute of Technology, Cambridge, Massachusetts, United States of America, 7 Mechanical and Industrial Engineering Department, Toronto Metropolitan University, Toronto, Ontario, Canada, 8 Dahdaleh Institute for Global Health Research, York University, Toronto, Canada, 9 Disaster & Emergency Management, York University, Toronto, Canada, 10 York Emergency Mitigation, Response, Engagement and Governance Institute, York University, Toronto, Ontario, Canada

¤ Current address: Laboratory of Industrial and Applied Mathematics (LIAM), York University, 8 The Chimneystack Rd, Toronto, Ontario, Canada
* grunnill@yorku.ca, m.d.grunnill@gmail.com (MG); wujh@yorku.ca (JW)

**Data Availability Statement:** Code used in simulations can be found at https://github.com/LIAM-COVID-19-Forecasting/Modelling-Disease-

## Abstract

The 2022 FIFA World Cup was the first major multi-continental sporting Mass Gathering Event (MGE) of the post COVID-19 era to allow foreign spectators. Such large-scale MGEs can potentially lead to outbreaks of infectious disease and contribute to the global dissemination of such pathogens. Here we adapt previous work and create a generalisable model framework for assessing the use of disease control strategies at such events, in terms of reducing infections and hospitalisations. This framework utilises a combination of meta-populations based on clusters of people and their vaccination status, Ordinary Differential Equation integration between fixed time events, and Latin Hypercube sampling. We use the FIFA 2022 World Cup as a case study for this framework (modelling each match as independent 7 day MGEs). Pre-travel screenings of visitors were found to have little effect in reducing COVID-19 infections and hospitalisations. With pre-match screenings of spectators and match staff being more effective. Rapid Antigen (RA) screenings 0.5 days before match day performed similarly to RT-PCR screenings 1.5 days before match day. Combinations of pre-travel and pre-match testing led to improvements. However, a policy of ensuring that all visitors had a COVID-19 vaccination (second or booster dose) within a few months before departure proved to be much more efficacious. The State of Qatar abandoned all COVID-19 related travel testing and vaccination requirements over the period of the World Cup. Our work suggests that the State of Qatar may have been correct in abandoning the pre-travel testing of visitors. However, there was a spike in COVID-19 cases and hospitalisations within Qatar over the World Cup. Given our findings and the spike in cases, we suggest a

Mitigation-at-Mass-Gatherings-A-Case-Study-of-
COVID-19-at-the-2022-FIFA-World-Cup.git.

**Funding:** MG position was funded through the
Fields Institute's Mathematics for Public Health
Next Generation program (http://www.fields.
utoronto.ca/activities/public-health), grant number
72062654. JA is funded through the Discovery
Grant program from the Natural Science and
Engineering Research Council of Canada (NSERC,
https://www.nserc-crsng.gc.ca/index_eng.asp),
grant number RGPIN-2017-05466. LB work is
supported, in part, by the US National Science
Foundation (NSF, https://www.nsf.gov/). AA is
funded through the Advanced Disaster, Emergency
and Rapid Response Simulation Initiative
(ADERSIM), Ontario Research Fund (https://www.
ontario.ca/page/ontario-research-fund) 33270. JW
work is also supported by the ADERSIM (Ontario
Research Fund 33270), along with the Canada
Research Chairs program (https://www.chairs-
chaires.gc.ca/home-accueil-eng.aspx, 230720),
and the Discovery Grant program from NSERC
(105588). This work was supported by the NSERC-
Sanofi Industrial Research Chair program in
Vaccine Mathematics, Modelling, and
Manufacturing (517504). The funders had no role
in study design, data collection and analysis,
decision to publish, or preparation of the
manuscript.

**Competing interests:** I have read the journal's
policy and the authors of this manuscript have the
following competing interests: LC is a Sanofi
employee and may hold stock options within
Sanofi. EWT is a Sanofi employee and may hold
stock options within Sanofi. AA is a Sanofi
employee and may hold stock options within
Sanofi.

policy requiring visitors to have had a recent COVID-19 vaccination should have been in place to reduce cases and hospitalisations.

## Author summary

Mass Gathering Events (MGEs) can potentially lead to outbreaks of infectious disease and facilitate the dissemination of such pathogens. We have adapted previous work to create a framework for simulating disease transmission and mitigation at such MGEs. We use the 2022 FIFA World Cup as a test case for this framework (modelling each match as independent 7 day MGEs). A policy of Pre-travel screenings of visitors was found to have little effect in reducing COVID-19 cases and hospitalisations. Pre-match screenings of spectators and match staff was found to be more effective. The most effective policy was to ensure that all visitors had a COVID-19 vaccination (second or booster dose) within a few months before departure. Qatar abandoned all COVID-19 related travel testing and vaccination requirements over the period of the World Cup. Our work suggests that the State of Qatar may have been correct in abandoning the pre-travel testing of visitors. However, there was a spike in COVID-19 cases and hospitalisations within Qatar over the World Cup. Given our findings and the spike in cases, we suggest a policy requiring visitors to have had a recent COVID-19 vaccination should have been in place to reduce cases and hospitalisations.

## 1 Introduction

The continuing COVID-19 pandemic, caused by an emerging coronavirus [1], has been affecting more than 200 countries since early 2020, profoundly overwhelming healthcare infrastructure worldwide [2, 3]. Given the initial lack of availability of effective drugs and vaccines, in order to control and contain the pandemic, governments and authorities have implemented a package of public health interventions. Such interventions have collectively become known as NPIs (Non-Pharmaceutical Interventions) [4]. In some countries, NPIs have included the ban of inter-household mingling and/or outdoor activities, particularly Mass Gathering Events (MGEs). As such, there is a large body of work pointing to the ban of MGE as an effective NPI [4]. Furthermore, there are several examples of outbreaks of communicable diseases at MGEs occurring in the pre-COVID era, many of which contributed to the global dissemination of the pathogens responsible [5–11]. The WHO defines MGEs as highly visible events attended by tens of thousands of people, such as pilgrimages and sporting events, and coordinates with member states on matters of pathogen control at such gatherings [12].

The resulting ban of MGEs in the wake of COVID-19 has affected the sporting world. Athletes have had to cope with unprecedented disruption, characterized by the loss of regular routine (e.g. training and matches), and the postponement or even cancellation of major national and international sporting events (Tokyo 2020 Olympic and Paralympic Games). During the pandemic, sports organizations have collaborated closely with the WHO and national public health bodies, assessing and implementing COVID-19-related risk reduction interventions to facilitate a safe step-wise return of sporting events [13]. Generally, these measures have focused on three areas. First, lowering the risk from the actual sport itself: focusing on activities that can maintain physical distancing; holding matches outdoors; re-consider whether to allow contact sports [13]. Second, lowering risks inherent to the size of an event, considering both

participants and spectators. Third, reducing risks inherent to the geographic localisation of the event by considering the local epidemiological conditions such as COVID-19 community transmission and prevalence. There are many COVID-19 mitigation protocols that can be utilised in planning for MGEs, either sporting events or others. However, the effectiveness and performance of these protocols in controlling and reducing the risk of COVID-19 transmission and hospitalisations is not clearly established.

Here we build on previous work [14] to create a generalised framework simulating disease transmission specifically adapted for use in planning pathogen control at MGEs. The 2022 FIFA World Cup hosted in Qatar was the first multi-continental sporting MGE of the post COVID-19 era to allow foreign spectators [15–17]. Denhing *et al.* (2023) and Subedi *et al.* (2022) [18, 19] highlighted the potential for disease spread at the world cup. Therefore, we chose to use the 2022 FIFA World Cup as a test case of the framework we developed. Modelling each match as independent 7 day MGEs, we assess various strategies to mitigate COVID-19 spread through match attendee testing and visitor vaccination requirements.

## 2 Methods

### 2.1 A generalised framework for simulating disease transmission at mass gatherings

In order to model the spread of COVID-19 at MGEs we have built upon our previous work [14] and created a generalised deterministic model framework (see Fig 1, Eq 1 and Tables 1 to 3). The general framework is that of a metapopulation stratified by clusters and vaccination groups, designated by subscripts *i* and *v*, respectively. Cluster composition is customisable to a range of MGEs. Specifically for the model outlined in this manuscript there are three main sets of clusters, the hosts, the visitor fans of team A and the visitor fans of team B. The effects of vaccination are controlled through parameters designated with a subscript *v*, effecting classes

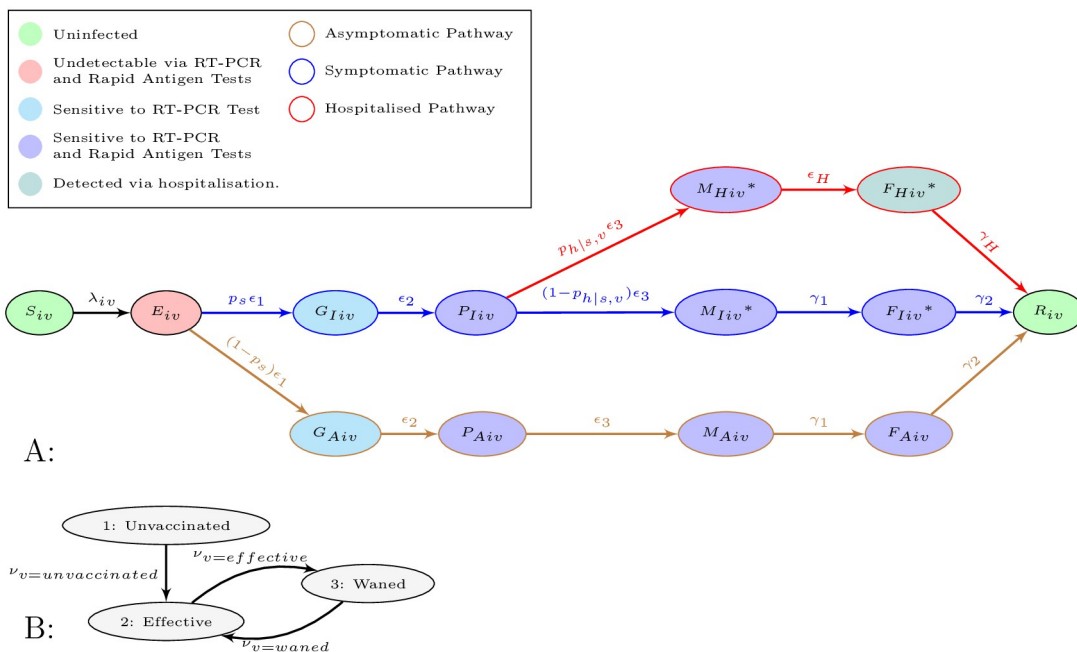

**Fig 1. Flow diagram of Model Classes (A) and Vaccination Groups (B).** A: all but the states with a * notation move between the vaccination groups depicted B at rates $\nu_{v=unvaccinated}$, $\nu_{v=effective}$ or $\nu_{v=waned}$.

**Table 1. Description of State Variables.**

| Variables | Descriptions |
|---|---|
| | Start of all Pathways |
| $S_{iv}$ | Susceptible population in cluster $i$ and vaccination group $v$. |
| $E_{iv}$ | Early latent infected population in cluster $i$ and vaccination group $v$. **Note** undetectable via RT-PCR and Rapid Antigen (RA) tests. |
| | Asymptomatic Pathway |
| $G_{Aiv}$ | Latent infected population in cluster $i$ and vaccination group $v$. **Note** Now detectable via RT-PCR. |
| $P_{Aiv}$ | Incubating infected population in cluster $i$ and vaccination group $v$. **Note** Now detectable via RA tests. |
| $M_{Aiv}$ | Mid-stage of infection population in cluster $i$ and vaccination group $v$. |
| $F_{Aiv}$ | Final Stage of infection population in cluster $i$ and vaccination group $v$. |
| | Symptomatic Pathway |
| $G_{Iiv}$ | Latent infected population in cluster $i$ and vaccination group $v$. **Note** Now detectable via RT-PCR. |
| $P_{Iiv}$ | Incubating infected population in cluster $i$ and vaccination group $v$. **Note** Now detectable via RA tests. |
| $M_{Iiv}$ | Mid-stage of infection population in cluster $i$ and vaccination group $v$. **Note** Now displaying symptoms. |
| $F_{Iiv}$ | Final Stage of infection population in cluster $i$ and vaccination group $v$. |
| | Hospitalised Pathway |
| $M_{Hiv}$ | Mid-stage of infection population in cluster $i$ and vaccination group $v$. **Note** Now displaying symptoms, but not yet hospitalised. |
| $F_{Hiv}$ | Hospitalised population in cluster $i$ and vaccination group $v$. |
| | End of all Pathways |
| $R_{iv}$ | Recovered population in cluster $i$ and vaccination group $v$. |

denoted within that vaccine group (see Vaccination groups). ODE integration of this model framework is achieved through Scipy's ODEint function [20].

$$
\begin{aligned}
dS_{iv}/dt = & \ v_{v-1}S_{iv-1} - \lambda_{iv}S_{iv} - v_v S_{iv} \\
dE_{iv}/dt = & \ v_{v-1}E_{iv-1} + \lambda_{iv}S_{iv} - (\epsilon_1 + v_v)E_{iv} \\
dG_{Aiv}/dt = & \ v_{v-1}G_{Aiv-1} + \epsilon_1(1 - p_s)E_{iv} - (\epsilon_2 + v_v)G_{Aiv} \\
dG_{Iiv}/dt = & \ v_{v-1}G_{Iiv-1} + \epsilon_1 p_s E_{iv} - (\epsilon_2 + v_v)G_{Iiv} \\
dP_{Aiv}/dt = & \ v_{v-1}P_{Aiv-1} + \epsilon_2 G_{Aiv} - (\epsilon_3 + v_v)P_{Aiv} \\
dP_{Iiv}/dt = & \ v_{v-1}P_{Iiv-1} + \epsilon_2 G_{Iiv} - (\epsilon_3 + v_v)P_{Iiv} \\
dM_{Aiv}/dt = & \ v_{v-1}M_{Aiv-1} + \epsilon_3 P_{Aiv} - (\gamma_1 + v_v)M_{Aiv} \\
dM_{Iiv}/dt = & \ (1 - p_{h|s,v})\epsilon_3 P_{Iiv} - \gamma_1 M_{Iiv} \\
dM_{Hiv}/dt = & \ p_{h|s,v}\epsilon_3 P_{Iiv} - \epsilon_H M_{Hiv} \\
dF_{Aiv}/dt = & \ v_{v-1}F_{Aiv-1} + \gamma_1 M_{Aiv} - (\gamma_2 + v_v)F_{Aiv} \\
dF_{Iiv}/dt = & \ \gamma_1 M_{Iiv} - \gamma_2 F_{Iiv} \\
dF_{Hiv}/dt = & \ \epsilon_H M_{Hiv} - \gamma_H F_{Hiv} \\
dR_{iv}/dt = & \ v_{v-1}R_{iv-1} + \gamma_2(F_{Aiv} + F_{Iiv}) + \gamma_H F_{Hiv} - v_v R_{iv}
\end{aligned}
\tag{1}
$$

**2.1.1 Disease stages.** Within each population cluster and vaccination group susceptible individuals, $S_{iv}$, can be infected through the force of infection $\lambda_{iv}$ (see Fig 1A, Eq 1 and Force of infection). Infection leads to the early latent stages of infection $E_{iv}$, where infection is not detectable through RT-PCR or Rapid Antigen (RA) tests. From here infected individuals progress ($\epsilon_1$) to one of two later latent phases $G_{Iiv}$ or $G_{Aiv}$, where infection is detectable through RT-PCR tests but not RA tests. Here an individuals infection pathway diverges either down a path leading to eventual symptoms at a proportion $p_s$ or asymptomatic infection at a proportion $1 - p_s$ (with classes denoted with subscripts $I$ and $A$, respectively) (see Fig 1A).

Infections become both transmissible, and detectable through RA tests, at rate $\epsilon_2$, moving to the incubating phases $P_{Iiv}$ and $P_{Aiv}$ [24, 41, 42]. From this stage on the asymptomatic track, $P_{Aiv}$, people progress at rate $\epsilon_3$ to stages $M_{Aiv}$, then at rate $\gamma_1$ to $F_{Aiv}$, finally recovering at rate $\gamma_2$ to $R_{iv}$. If on the symptomatic track $P_{Iiv}$ people progress to the first stages of symptoms at rate $\epsilon_3$. Here there is a risk of people progressing down the hospitalisation pathway, moving to stage $M_{Hiv}$, at probability $p_{h|s,\,v}$ (see Fig 1A). Eventually individuals in $M_{Hiv}$ are hospitalised at rate $\epsilon_H$ moving to compartment $F_{Hiv}$. It is assumed that those hospitalised do not contribute to the force of infection (see Force of infection). Recovery from hospitalisation, $F_{Hiv}$, occurs at rate $\gamma_H$ and leads to stage $R_{iv}$. If a person does not move to the hospitalised pathway, $1 - p_{h|s,\,v}$, they remain on the symptomatic pathway develop symptoms and progress to stage $M_{Iiv}$. From here people progress to the final stage of infection $F_{Iiv}$ at rate $\gamma_1$ and then to recovered class, $R_{iv}$, at rate $\gamma_2$.

**2.1.2 Vaccination groups.** All individuals start in the Unvaccinated group (indexed as 1 in Fig 1B). After completing a primary series of vaccination people move to the Effective vaccination group, $v_{v=unvaccinated}$ (indexed as 2 in Fig 1B). Several months after primary series of vaccination immunity wanes [26, 43] moving people from vaccine group Effective to Waned, $v_{v=effective}$. The waned vaccination group is indexed as group 3. Note subscript v indexes the vaccination group not the number of doses of a vaccine. Individuals in the Waned vaccination group can receive a booster dose, at rate $v_{v=waned}$, moving them back to the Effective vaccination group. Again after several months in the effectively vaccinated group immunity wanes, at rate $v_{v=effective}$, moving people to the Waned vaccination group. In other words, after a primary series of vaccination people loop from the Effective vaccination group to the Waned vaccination group through the waning of immunity, $v_{v=effective}$, and back again with booster doses, $v_{v=waned}$. In concordance with many national public health agencies' advice [44, 45] only non-symptomatic people (i.e. all classes but $M_{Iiv}$, $M_{Hiv}$, $F_{Iiv}$, and $F_{Hiv}$) can be vaccinated at rates $v_{v=unvaccinated}$ or $v_{v=waned}$. The effectiveness of vaccination plays out in the different vaccine groups, through modification of force of infection ($\lambda_{iv}$) and hospitalisation ($p_{h|s,v}$), see Eq 2, Tables 2 and 3

$$
\begin{aligned}
\lambda_{iv} &= \lambda_i(1 - l_v) \\
p_{h|s,v} &= p_{h|s}(1 - h_v)
\end{aligned}
\tag{2}
$$

**2.1.3 Clusters.** Clusters come under two main categories, visitor clusters and host clusters (see Table 4). In order to simulate COVID-19 screening, each of these main clusters have associated clusters for "RA Positive", "Waiting for Positive RTPCR" and "RTPCR Positive". Tests are simulated using the event queue system (see Event queue). In the case of RA test events, a proportion of a clusters population ($\tau_{RA}$) from states $P_{Iiv}$, $M_{Iiv}$, $F_{Iiv}$, $M_{Hiv}$, $P_{Aiv}$, $M_{Aiv}$ and $F_{Aiv}$ are moved to the associated RA Positive cluster (see Fig 1A). This detected proportion being based on the RA tests sensitivity [41]. Those in the RA Positive cluster are isolating and thereby contribute less to transmission (see Table 4 and Force of infection). RT-PCR tests are capable

**Table 2. Description of Parameters Volume 1: Disease Progression.**

| Parameters | Descriptions | Values | Sources |
|---|---|---|---|
| $p_s$ | Probability of developing symptoms. | 0.41 to 0.84 | [21–23] |
| $\epsilon_1$ | Progression from early latent phase **and** to being RT-PCR detectable. | 0.598 day$^{-1}$ | [24] |
| $\epsilon_2$ | Progression from later latent phase to incubating infection phase (infectious) **and** being Rapid Antigen (RA) test detectable. | 1 day$^{-1}$ | [24] |
| $\epsilon_3$ | Progression from incubating phase to mid-infection phase. If not on asymptomatic pathways this is also progression to displaying symptoms. | 1 day$^{-1}$ | [24] |
| $p_h$ | Probability of being hospitalised if unvaccinated. | 0.0 to 0.0234 | Upper bound is from [25]. Lower bound assumes decreasing morbidity with future strains. |
| $p_{h\mid s}$ | Probability of being hospitalised given symptoms if unvaccinated. | $\frac{p_h}{p_s}$ | |
| $h'_{v=effective}$ | Efficacy of vaccination with regards to hospitalisation for those effectively vaccinated | 0.837 to 1 | Range in vaccine effectiveness against infection leading to hospitalisation or death seen over the first 6 months since second dose [26] |
| $h'_{v=waned}$ | Efficacy of vaccination with regards to hospitalisation for the waned vaccination group | 0.5560 | Vaccine effectiveness against infection leading to hospitalisation or death after 6 months from second dose [26] |
| $h_v$ | Efficacy of vaccination with regards to hospitalisation given efficacy against infection for vaccination group $v$. | $1 - \frac{1-h'_v}{1-l_v}$ | Transformation taken from [27, 28]. |
| $p_{h\mid s,v}$ | Probability of being hospitalised in vaccination group $v$. | $p_{h\mid s}(1h_v)$ | |
| $\gamma^{-1}$ | Total time infected for symptomatic or asymptomatic pathway. | 10 days | [24] |
| $\gamma_1$ | Progression from mid asymptomatic and symptomatic infection to late stage infection. | $\frac{2}{\gamma^{-1}-\epsilon_1^{-1}-\epsilon_2^{-1}-\epsilon_3^{-1}}$ | |
| $\gamma_2$ | Recovery from final phase of asymptomatic or symptomatic infection. | $\frac{2}{\gamma^{-1}-\epsilon_1^{-1}-\epsilon_2^{-1}-\epsilon_3^{-1}}$ | |
| $\epsilon_H$ | Rate of hospitalisation. | 0.103 to 0.3820 day$^{-1}$ | [29–31] |
| $\gamma_H$ | Recovery from hospitalisation. | 0.0448 to 0.1550 day$^{-1}$ | [32] |

Note parameters are sampled from a uniform distribution via Latin Hypercube sampling, if not fixed.

of detecting the presence of COVID-19 earlier in an infection [41], meaning that the proportion of a clusters population ($\tau_{RT-PCR}$) is also drawn from states $G_{Iiv}$ and $G_{Aiv}$ (see Fig 1A). However, RT-PCR tests have a much longer turnaround time [41], typically a day or two [38–40]. Therefore, the detected proportion from RT-PCR tests ($\tau_{RT-PCR}$) will populate a "Waiting for Positive RT-PCR" cluster. All the classes in the "Waiting for Positive RT-PCR" cluster transition to the associated "RTPCR Positive" cluster at rate $\omega_{RT-PCR}$ (RTPCR turnaround time). As with the RA Positive cluster, the RT-PCR Positive cluster is isolating and thereby contributes less to transmission (see Table 4 and Force of infection).

**2.1.4 Event queue.** In order to simulate changes in parameter values (such as increasing transmission) and the transfer of population between compartments (e.g. moving to isolation) an event queue system has been employed. This runs a model between events, then changes a parameter value, adds or deducts from compartments in a compartment model depending on the event. The code for this has been made freely available (see link in the data availability statement). A note of caution with making comparisons between scenarios with events at different times. If no event occurs at a time point in one scenario but there is an event in the other at that time, a null (do nothing) event must be inserted at that time point for simulations made without the event at that time. This is critical to ensure comparable accuracy of the integration for simulations of distinct scenarios.

**Table 3. Description of Parameters Volume 2: Force of Infection and Testing.**

| Parameters | Descriptions | Values | Sources |
|---|---|---|---|
| $l_v$ | Vaccine effectiveness against infection for those in vaccine group v. | | |
| $l_{v=effective}$ | Vaccine effectiveness against infection for those effectively vaccinated. | 0.2230 to 0.7750 | Range in vaccine effectiveness against infection seen over the first 6 months since second dose [26]. Note lower bound has been truncated from 0.1730 to 0.2230, this avoids sampling from parameter space that suggests an increase in vaccine efficacy after 6 months. |
| $l_{v=waned}$ | Vaccine effectiveness against infection for the waned vaccination group. | 0.2230 | Vaccine effectiveness against infection after 6 months from second dose [26] |
| $\lambda_i$ | Force of infection experienced by cluster $i$. | person$^{-1}$ day$^{-1}$ | |
| $\lambda_{iv}$ | Force of infection experienced by cluster $i$ and vaccination group $v$. | $\lambda_i(1 - l_v)$ person$^{-1}$ day$^{-1}$ | |
| $\theta$ | Modification of transmission from asymptomatic and pre-symptomatic states. | 0.3420 to 1 person$^{-1}$ day$^{-1}$ | Lower bound from [21, 33, 34]. Upper bound assumes no difference in transmission from symptomatic states. |
| $\kappa$ | Modification in transmission due to quarantine/isolation as those in this cluster ($i$) have tested positive via RT-PCR or RA test. | 0 to 1 person$^{-1}$ day$^{-1}$ | Covers assumptions of completely successful (0) to completely unsuccessful isolation (1). |
| $R_0$ | Basic reproduction number for a single cluster (homogeneous mixing) and no vaccination. | 2 to 10 | Covers range seen in [29] and median estimate for Omicron strain [35]. |
| $\beta$ | Baseline transmission from infectious states. | Derived from $R_0$ (person$^{-1}$ day$^{-1}$) | See S1 Methods |
| $b$ | Increase in transmission for those that attend the sports match (day 3). | 1 to 78.5 person$^{-1}$ day$^{-1}$ | Lower bound assumes no increase. Upper bound taken to be increase in meningococcal transmission seen with Hajj [14]. |
| $\nu_v$ | Rate of progress from one vaccination group to the next (e.g. any arrow in Fig 1B). | 0 day$^{-1}$ | |
| $\nu_{v=unvaccinated}$ | Rate of completing primary vaccination series. | 0 day$^{-1}$ | |
| $\nu_{v=effective}$ | Rate of waning immunity of vaccination. | 0 day$^{-1}$ | |
| $\nu_{v=waned}$ | Rate of receiving a booster vaccination (not necessarily first booster vaccination). | 0 day$^{-1}$ | |
| $\tau_{RA}$ mid | RA test sensitivity mid value. | 0.728 test$^{-1}$ | Estimate from [36] |
| $\tau_{RA}$ low | RA test sensitivity low value. | 0.624 test$^{-1}$ | Ricco *et al.,* 2022 [36] acknowledges that $\tau_{RA}$ mid is likely an overestimate, therefore we compare simulations using $\tau_{RA}$ mid with the lower confidence interval from Ricco *et al.,* 2022 [36]. |
| $\tau_{RT-PCR}$ | RT-PCR test sensitivity. | 0.968 test$^{-1}$ | [37] |
| $\omega_{RT-PCR}$ | RT-PCR test turnaround time. | 1 day$^{-1}$ | Turnaround time seen other mass testing regimes [38–40] |

Note parameters are sampled from a uniform distribution via Latin Hypercube sampling, if not fixed.

**2.1.5 Force of infection.** Force of infection is calculated for each cluster summing up the contribution from all clusters (including itself) ($j$) and their vaccination groups ($v$) (see Eq 3). As already mentioned states that do not display symptoms have their transmission modified by $\theta$. Isolation is achieved in "RA Positive" and "RTPCR Positive" clusters by their $\kappa_j = \kappa$, for other clusters $\kappa_j = 1$.

$$\lambda_i = \sum_{j=1}^{n_j} \frac{\kappa_j \beta_{ij}(\sum_{v=1}^{n_v} \theta(P_{Ijv} + P_{Ajv} + M_{Ajv} + F_{Ajv}) + M_{Ijv} + F_{Ijv} + M_{Hjv})}{N_{i*}} \tag{3}$$

The transmission term $\beta_{ij}$ refers to transmission to cluster $i$ from cluster $j$. For the majority of simulation time this is set at a baseline ($\beta_{ij} = \beta$). However, this can be changed using the event queue system to have $\beta_{ij} = b\beta$ for a period of time, $b$ being a strengthening or weakening of transmission over that time period. $N_{i*}$ represents the population in which the

**Table 4. Description of Cluster Behaviour and Organisation.**

| Name | Host or Visitor | Attends Match | Isolating |
|---|---|---|---|
| Host | Host | | |
| Host: Positive RA | Host | | ✓ |
| Host: Waiting for Positive RTPCR | Host | | |
| Host: Positive RTPCR | Host | | ✓ |
| Host Spectators | Host | ✓ | |
| Host Spectators: Positive RA | Host | | ✓ |
| Host Spectators: Waiting for Positive RTPCR | Host | ✓ | |
| Host Spectators: Positive RTPCR | Host | | ✓ |
| Host Staff | Host | ✓ | |
| Host Staff: Positive RA | Host | | ✓ |
| Host Staff: Waiting for Positive RTPCR | Host | ✓ | |
| Host Staff: Positive RTPCR | Host | | ✓ |
| Team A Fans | Visitor | ✓ | |
| Team A Fans: Positive RA | Visitor | | ✓ |
| Team A Fans: Waiting for Positive RTPCR | Visitor | ✓ | |
| Team A Fans: Positive RTPCR | Visitor | | ✓ |
| Team B Fans | Visitor | ✓ | |
| Team B Fans: Positive RA | Visitor | | ✓ |
| Team B Fans: Waiting for Positive RTPCR | Visitor | ✓ | |
| Team B Fans: Positive RTPCR | Visitor | | ✓ |

interaction between a susceptible individual of cluster $i$ ($S_{iv}$) and an infectious individual of cluster $j$ ($P_{Ijv}$, $P_{Ajv}$, $M_{Ajv}$, $F_{Ajv}$, $M_{Ijv}$, $F_{Ijv}$, or $M_{Hjv}$) takes place. Similarly to the transmission term ($\beta_{ij}$), $N_{i*}$ is typically set at the baseline value of the entire population being modelled ($N$). However, this can be changed using the event queue system allowing for transmission to be modelled through interactions taking place within certain sub-populations. Note the summation term $\sum_{v=1}^{n_v}$ means to sum through all the infectious stages of all the vaccination groups of cluster j, in this case vaccine groups 1: Unvaccinated, 2: Effective and 3: Waned. Recall from Vaccination groups that the subscript $v$ indexes the vaccination group not the number of doses of a vaccine.

## 2.2 Simulating FIFA 2022 World Cup matches, as a case study

For a test case scenario of the generalised framework above (see A generalised framework for simulating disease transmission at mass gatherings), we chose to model possible matches from the FIFA 2022 World Cup (not involving the Qatari team). Each match is seen as an entirely independent 7 day MGE (see Uncertainty and sensitivity analyses), we do not model the FIFA 2022 World Cup as a whole.

**2.2.1 Simulation of a FIFA 2022 World Cup match.** For each match there were five main clusters, one for hosts in general, one for host spectators, one for host staff and two clusters of visitor fans, one for each team, (see Tables 4 and 5). The eight stadiums hosting matches have estimated capacities ranging from 40,000 to 80,000 [46]. As a means of exploring parameter uncertainty, we assume therefore that the population attending simulated fixtures ranges from 4,000 to 80,000 ($N_A$). A proportion of tickets go to the host spectator cluster ($0 < = \eta_{spectators} < = 0.5$), meaning that the two visitor clusters made up the rest of the attendees, $N_A$, split evenly. The host staff cluster population, $N_S$, ranged from 4,000 to 20,000. The

**Table 5. Starting Values of Variables used for Simulating a FIFA 2022 World Cup Match.**

| Variables | Descriptions | Values | Sources |
|---|---|---|---|
| $N_{hosts}$ | Combined population of host clusters (population of Qatar) | 2930524 people | [47] |
| $N_{hosts,full}$ | Combined fully vaccinated population for all host clusters (e.g. $N_{hosts,eff} + N_{hosts,wan}$) | 2848639 people | Qatari Fully Vaccinated population for 15/11/2022 [48]. |
| $N_{hosts,eff}$ | Effectively vaccinated population across all host clusters. | 1898869 people | Qatari Booster vaccines given for 15/11/2022 [48] |
| $N_A$ | Population of attendees of sports match. | 4,000 to 80,000 people | Lower bound assumes a tenth of the tickets of the lowest capacity stadium are sold [46]. Upper bound is the capacity of the largest stadium [46]. |
| $N_Q^*$ | Proportion of tickets given to host population. 0 to 0.5 | | |
| $N_S$ | Population of hosts staffing sports match. | 4,000 to 20,000 people | A tenth of the typical stadium capacity to a quarter of the maximum stadium capacity [46]. |
| $\sigma_H$ | Prevalence in host nation. | 0.0006 to 0.0011 person$^{-1}$ | Inverse of Uncertainty Intervals for Qatari cumulative detection to infection ratio in [49] multiplied by Qatar's prevalence 18/11/2022 [48]. |
| $\sigma_A$ and $\sigma_B$ | Prevalence in nation A and B, respectively. | $4.47 \times 10^{-6}$ to 0.0030 person$^{-1}$ | Inverse of the maximum and minimum of Uncertainty Intervals for cumulative detection to infection ratio of non-Qatari teams playing at FIFA World Cup2022 [49] multiplied by non-Qatari sides prevalence 18/11/2022 [48]. |
| $v_A$ and $v_B$ | Proportion effectively vaccinated arriving from nations A and B, respectively. The rest of the visitors are in the waned vaccinations group. | 0 to 1 | |

Note parameters are sampled from a uniform distribution via Latin Hypercube sampling, if not fixed.

host general population cluster equaled the population of Qatar, 2,930,524 [47], minus the host spectator ($N_A * \eta_s$) and staff cluster ($N_S$) populations.

Vaccination and waning of immunity are not considered during these simulations ($v_{v=unvaccinated} = v_{v=waned} = v_{v=effective} = 0$), as the simulations occur over a short time frame. The host unvaccinated population was set at Qatar's population minus the number of people fully vaccinated in Qatar as of 15/11/2022 [48]. The hosts effectively vaccinated population was set as the total boosters given as of 15/11/2022 [48]. The hosts waned vaccination group was populated with people fully vaccinated minus total boosters given.

Team A and B fans were assumed to have at least completed a primary series of vaccination. Prior to the world cup Qatar had travel restrictions requiring a primary series of vaccination to access public facilities [17, 50, 51]. The proportion of effectively vaccinated in these two clusters therefore ranged between simulations, $0 <= v_A <= 1$ for Team A and $0 <= v_B <= 1$ for Team B fans. The remaining population of these two clusters was placed in the waned vaccination group.

The Host, Host Spectator, Host Staff, Team A and Team B Clusters were seeded with COVID-19 infections. For the three host clusters the starting prevalence, $\sigma_H$, was sampled from a range (see Table 5). The 7-day smoothed new cases per person for Qatar on 18/11/2022 [47, 48], multiplied by the lower and upper estimate of reported to actual infections for Qatar [49] to give this range. Similarly, the starting prevalence for Team A and B fans ($\sigma_A$ and $\sigma_B$) was also sampled from a range based on the smoothed new cases per person 18/11/2022 [47, 48]. The new smoothed cases per person for each nation was multiplied by the respective lower and upper estimate of reported to actual infections [49]. The minimum and maximum from this set of values then informed the range for starting prevalences for Team A and Team B fans (see Table 5).

We chose a probabilistic approach to pick the seed infection stages. First we made a random draw to select which infection pathway (branch) a host is on, using the probability of being on an infection pathway. Then each infection stage of a pathway was assigned a weight. The weight was calculated as the inverse of the outflow rate from that compartment. We normalised the weights for each infectious stage by dividing by the sum of all weights, prior to using the result to draw the selection of infection stage. All draws were made using numpy's multinomial function [52] see code linked in the data availability statement. The seed number used for these random number draws was generated as part of sampling parameter space, see Uncertainty and sensitivity analyses.

The baseline transmission term $\beta$ was derived for a given value of $R_0$, assuming no vaccination and a single cluster population. $R_0$ was derived using Next Generation Matrix Methods [53] and sympy [54] (see S1 Methods for details). Simulation then proceeded as outlined in Table 6. We initiated the simulation 2 days prior to the actual MGE so as to capture pre-travel COVID-19 screenings [55]. The simulation extended from 7 to 100 days post MGE without transmission, so as to capture the number of hospitalisations resulting from transmission during the MGE.

**2.2.2 Uncertainty and sensitivity analyses.** Parameters and starting variable values were either held fixed or sampled using Latin Hypercube Sampling (LHS) [56], using scipy's Latin-Hypercube function [20] (see Tables 2, 3 and 5). LH sampling was done using uniform distributions and a sample size of 10,000. In order to more accurately compare simulations made using the same sample of parameters, the seed number used to select the states of the initial infected population was generated within LH sampling, drawing from a uniform distribution (0 to 1,000,000,000). Partial Rank Correlation Coefficients (PRCCs) were then used to asses the effect of each sampled parameter on total hospitalised, peak hospitalised, total infected and peak infected. PRCCs were calculated using pingouin's partial_corr function [57].

**2.2.3 Analyses of testing strategies.** In order to asses the effect of different test strategies the same LH sample was run with each of the testing regimes described in Table 7. The effectiveness of the test strategies was measured through two sets of comparisons using the outputs total infections, peak infections, total hospitalisation and peak hospitalisation. The first set of comparisons were PRCC based. Each set of simulations made under a testing strategy was paired with the set of simulations made with no testing regime in place, as a control. For simulations under the test strategy a dummy parameter was given a value of 1. Simulations made without a testing regime in place were given a value of 0 for this dummy parameter. Thus, creating a parameter to base PRCC comparisons on. The second set of comparisons measured a testing regime's percentage relative differences in outputs, using Eq 4, compared to the "No Testing" regime as a control. Regarding Eq 4, $R_l$ is the percentage relative difference in an output $O$ seen between a simulation with a treatment $T$ and a control simulation $C$, where $l$ is the LH sample used in the two simulations being compared.

Ricco *et al.*, 2022 [36] acknowledges that their estimated value for RA sensitivity ($\tau_{RA}$ mid) is likely an overestimate. Therefore, simulations using RA tests were run twice once using $\tau_{RA}$ mid and the lower confidence interval from Ricco *et al.*, 2022 [36] ($\tau_{RA}$ low). Testing strategies using RA tests with the $\tau_{RA}$ mid are denoted as RA mid. Likewise, testing strategies using RA tests with the $\tau_{RA}$ low are denoted as RA low.

$$R_l = \frac{O_{Tl} - O_{Cl}}{O_{Cl}} \times 100 \tag{4}$$

**Table 6. Event Timeline used for Modelling International Sports Matches.**

| Time of Events | Events | Description |
|---|---|---|
| -2 | Simulation Begins | • Transmission to and from visitor clusters is 0.<br>• Transmission between host clusters is at baseline ($\beta$).<br>• For host clusters the population denominator for the force of infection is the population of Qatar. |
| -1.5 | Pre-Travel RTPCR or Null Event | • Pre-Travel RTPCR: a proportion of those in the RTPCR detectable states are removed ($\tau_{RTPCR}$) from visitor clusters. |
| -0.5 | Pre-Travel RA or Null Event | • Pre-Travel RA: a proportion of those in the RA detectable states are removed ($\tau_{RA}$) from visitor clusters. |
| 0 | MGE Begins and Visitor Clusters Arrive | • Transmission to and from visitor clusters is set to baseline ($\beta$).<br>• For all clusters the population denominator for the force of infection is set to the population of Qatar plus that of the two visitor clusters. |
| 1.5 | Pre-Match RTPCR or Null Event | • Pre-Match RTPCR: a proportion of those in the RTPCR detectable states are moved ($\tau_{RTPCR}$) from clusters attending the match to their associated "Waiting for Positive RTPCR" cluster. |
| 2.5 | Pre-Match RA or Null Event | • Pre-Match RA: a proportion of those in the RA detectable states are moved ($\tau_{RTPCR}$) from clusters attending the match to their associated "Positive RA" cluster. |
| 3 | Match Day Begins | • Transmission to and from match attending clusters is increased by a factor, $b$).<br>• For transmission to clusters attending the match the population denominator for the force of infection is set to the population attending the match.<br>• For transmission to clusters not attending the match the population denominator for the force of infection is set to the population not attending the match. |
| 4 | Match Day Ends | • All transmission terms to and from match attending clusters is reset to baseline $\beta$.<br>• The force of infection is set to the population the Qatari population plus visitors for all clusters. |
| 7 | MGE ends | • All transmission terms, $\beta_{ij}$, are set to 0. |
| 100 | Simulation Ends | |

**2.2.4 Analyses of travel vaccination restrictions.** The proportions of visitor A and B effectively vaccinated ($v_A$ and $v_B$) where both found to have significantly negative PRCCs with infections and hospitalisation (see Effects of parameters and starting conditions relating to COVID-19 control measures). This suggests that a policy restricting entry to those effectively

**Table 7. Testing Regimes Employed in Simulations.**

| Strategy | RT-PCR day -1.5 | RA day -0.5 | RT-PCR day 1.5 | RA day 2.5 |
|---|---|---|---|---|
| No Testing | | | | |
| Pre-Travel RT-PCR | ✓ | | | |
| Pre-Travel RA (low and mid) | | ✓ | | |
| Pre-Match RT-PCR | | | ✓ | |
| Pre-Match RA (low and mid) | | | | ✓ |
| Double RT-PCR | ✓ | | ✓ | |
| Double RA (low and mid) | | ✓ | | ✓ |
| RT-PCR then RA (low and mid) | ✓ | | | ✓ |
| RA (low and mid) then RT-PCR | | ✓ | ✓ | |

vaccinated but no COVID-19 screening being enforced, was worth evaluating. Henceforth we will refer to such a policy as "effective visitor vaccination". Therefore, a further LHS of size 10,000 was drawn, this time without $v_A$ and $v_B$ from Tables 2, 3 and 5 being sampled. The LHS parameter sets were then used to simulate a policy of "effective visitor vaccination" ($v_A = v_B = 1$ and no testing being in place). Calculations of percentage relative differences in total infections and hospitalisations between this policy, as a control, against simulations made under a different combination of testing regime and visitor effective vaccination ($v_A = v_B$) with the same LH sample set as treatments were made (see Eq 4). These combinations comprised of $v_A = v_B = 0$, $v_A = v_B = 0.25$, $v_A = v_B = 0.5$ or $v_A = v_B = 0.75$ with "No Testing", "Pre-Travel RT-PCR", "Pre-Match RT-PCR", "Pre-Match RA" or "RT-PCR then RA" testing regimes. Thereby, capturing a testing regime being in place with different background levels of visitor effective vaccination.

## 3 Results

Here we focus on an analysis of testing regimes, along with the parameters and starting conditions relating to COVID-19 control measures. S1 Results contain further analyses of the effects of other parameters and starting conditions that we varied through LHS. We note that for nearly all PRCCs of starting conditions, parameters and testing regimes against peak infections and hospitalisation followed the same trends as total infections and hospitalisation. The exceptions being rate of hospitalisation and recovery from hospitalisation. Similarly % relative differences caused by testing regimes follow the same trend when comparing the peak and total number of infections or hospitalisations (see S1 Results).

### 3.1 Effects of testing regimes

PRCCs from single testing regime showed much higher performance for pre-match testing over pre-travel testing in reducing infections and hospitalisations (see Fig 2). Furthermore, pre-travel screenings provide less reductions in infections and hospitalisation compared to pre-match screenings (see Fig 3). Single RT-PCR tests had a greater benefit in pre-travel testing, but perform similarly to RA tests in pre-match and double testing, when considering the two RA test sensitivity values used. Combination of pre-travel and pre-match test events lead to improvements in over single pre-match tests (see Figs 2 and 3). If outlier reductions in infections and hospitalisations are considered "RT-PCR then RA" testing regime (pre-travel RA and pre-match RA), is the best performing test (see Fig 3).

### 3.2 Effects of parameters and starting conditions relating to COVID-19 control measures

In terms of active control measures decreasing the transmission from isolating clusters would only be effective in testing regimes that included pre-match testing Fig 4. Note, pre-travel tests remove positive visitors from the model. However, greater reductions in infections and hospitalisations are seen through reduced transmission from pre-symptomatic and asymptomatic people. This could be achieved through many NPIs, such as encouraging or enforcing mask wearing, promoting hand sanitation and, when possible, social distancing.

The proportions of visitor clusters A and B effectively vaccinated ($v_A$ and $v_B$) have a negative correlation with both infections and hospitalisation (see Fig 4). Differences in PRCCs can be transformed to z-scores, as outlined in [56]. These methods were used to determine if the effects of Testing Regimes and $v_A$ and $v_B$ under the 'No Testing' regime are significantly different (compare Figs 2 and 4). PRCCs of $v_A$ and $v_B$ compared to single pre-travel screening testing regimes demonstrate a significantly greater effect in reducing hospitalisations and

infections (see one tailed p-values in S1 and S2 Tables). $v_A$ and $v_B$ have a significantly greater effect in reducing hospitalisations compared to testing regimes involving pre-match testing. However, such testing regimes had a significant but slightly superior effect in reducing infections compared to $v_A$ and $v_B$ (see one tailed p-values in S1 and S2 Tables).

### 3.3 Effects of proportion of recently vaccinated as a COVID-19 control measure

It can be seen from Figs 5 and 6, that "effective visitor vaccination" ($v_A = v_B = 1$ and no testing being in place) outperforms the "Pre-Travel RTPCR" testing regime, reducing both hospitalisations and infections. When it comes to the pre-match and "RT-PCR then RA" testing regimes, "effective visitor vaccination" outperforms for reductions in hospitalisations. At $v_A = v_B = 0.75$ the median difference in infections were slightly less than 0 but the

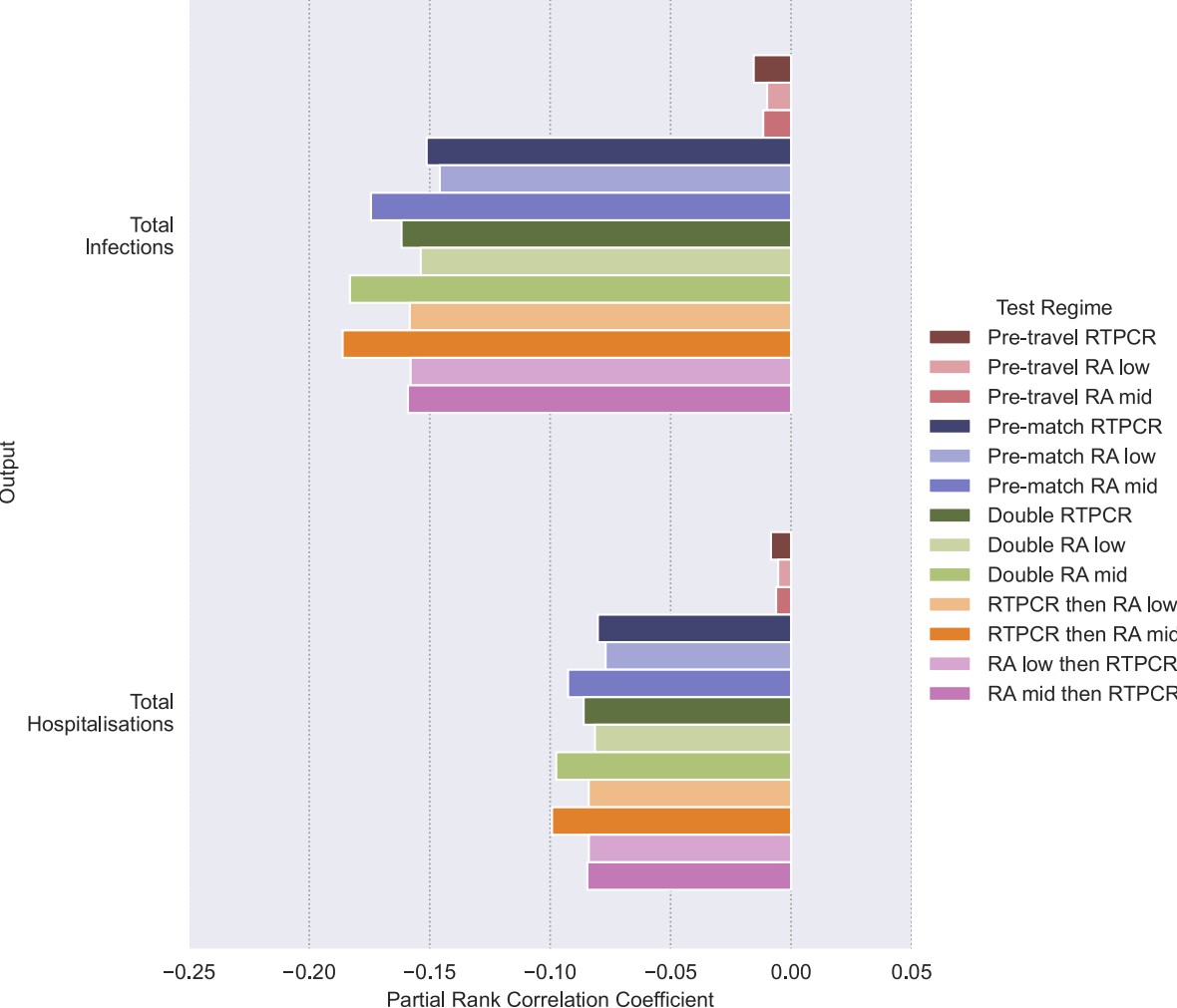

**Fig 2. Effect of different Test Regimes on infections and hospitalisations as measured by Partial Rank Correlation Coefficient (PRCC).** In calculating PRCCs Latin Hypercube (LH) sampling draws on the parameter space outlined in Tables 2, 3 and 5, using uniform distributions. Simulations are made with the resulting LH sample with each of the testing regimes outlined in Table 7. Every set of simulation made under a testing regime is given a dummy parameter value of 1, except "No Testing" which is given a value of 0. Each testing regime's effect on an output (Total Infections or Hospitalisation) is measured through calculating PRCCs using the dummy parameter comparing the 1 for the particular testing regime and 0 for its absence.

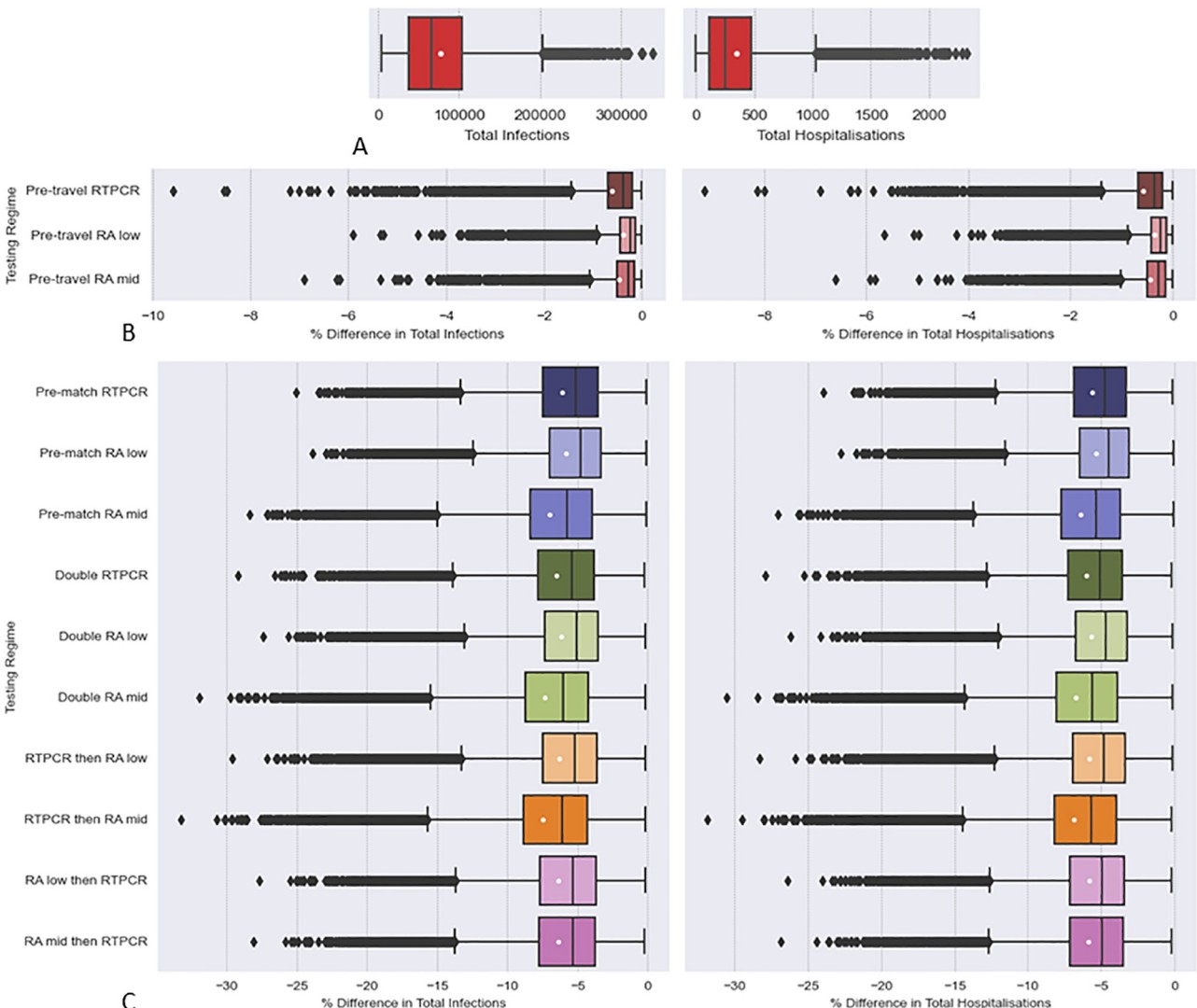

**Fig 3. Effect of different Test Regimes on infections and hospitalisations as measured by % Relative Difference to simulations with no testing regime.** A: Boxplots Total Infections and Hospitalisation in simulations made with no testing regime. B and C: Boxplots of a Testing Regimes % Relative Differences in Total infections and Hospitalisation. For every parameter set produced under LHS the % relative difference in outputs simulated under a testing regime, Fig 3B and 3C, was calculated against the corresponding output from the "No Testing" regime simulations, depicted in Fig 3A, as a control (see Eq 4). The white dots are the means. The array of samples used in simulation was generated from Latin Hypercube sampling drawing upon the distributions outlined in Tables 2, 3 and 5. Details of testing regimes can be found in Table 7.

mean was closer to 0 (or slightly above) for pre-match and "RT-PCR then RA" testing regimes. At $v_A = v_B = 0.5$ and $v_A = v_B = 0.25$ median % relative differences in infections are around (to slightly above) 0 for pre-match and "RT-PCR then RA" testing regimes, with mean values being above 0. At $v_A = v_B = 0.0$ both median and mean % relative differences in infections are better than 0. The outlier % relative differences in infections tend to be positive, with more being positive as the proportion of visitors effectively vaccinated increases. Overall, this suggests that an "effective visitor vaccination" policy generally leads to similar reductions in infections compared to pre-match and "RT-PCR then RA" testing regimes, but is more likely to be an improvement in select circumstances.

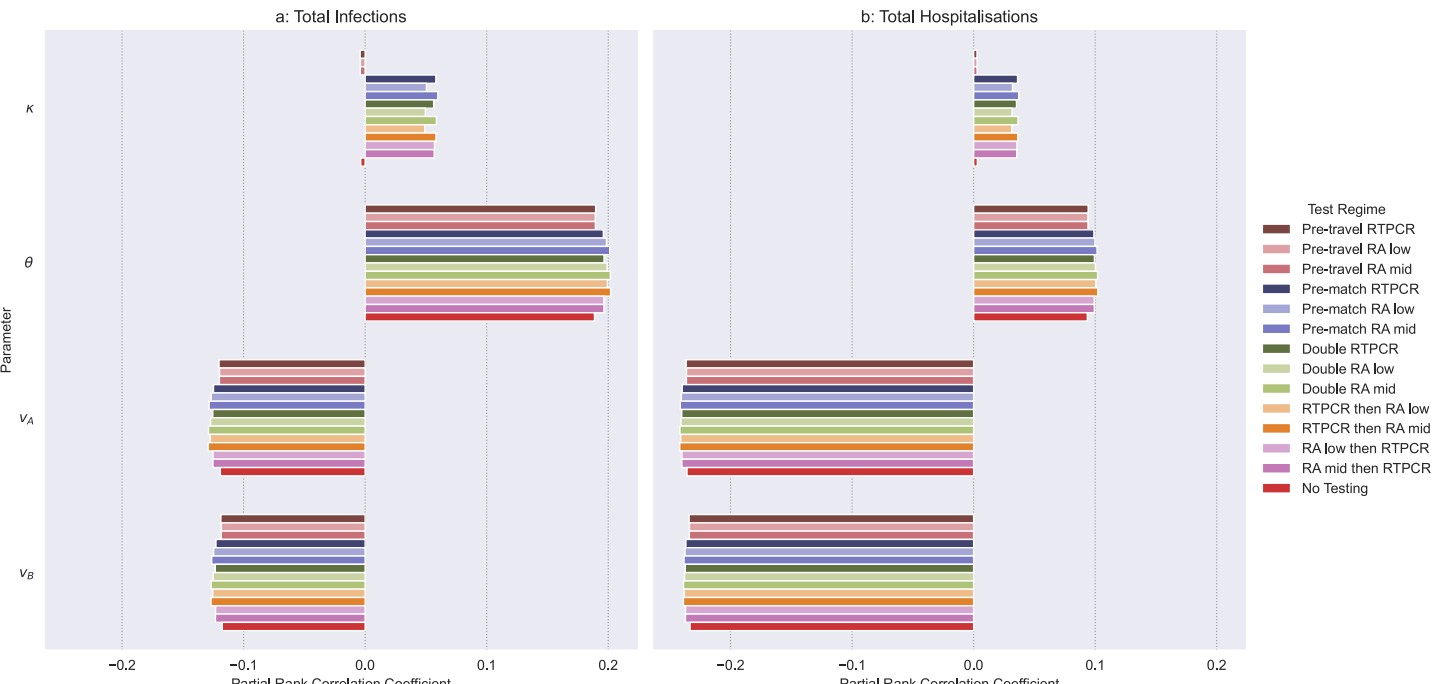

**Fig 4. Partial Rank Correlation Coefficients (PRCCs) between parameters and starting conditions relating to COVID-19 control measures and Total Infections and Hospitalisations.** Where, $\kappa$ is the isolation transmission modifier (0–1), $\theta$ is the asymptomatic transmission modifier (0.342–1), and $v_A$ and $v_B$ are the proportion recently vaccinated visitors in group clusters A and B, respectively, (0–1). The array of samples used in simulation was generated from Latin Hypercube sampling drawing upon the distributions outlined above and in Tables 2, 3 and 5, using uniform distributions. Details of testing regimes can be found in Table 7.

## 4 Discussion

Major MGEs such as religious pilgrimages, festivals or sport competitions can generate a variety of health risks. In the context of an ongoing infectious disease pandemic, in addition to risks at the host site, risks of amplification or the dissemination of the pathogen to regions from which it was originally absent or close to it. MGEs have the potential to enable or favor the evolution and spread of novel variants of SARS-CoV-2 and other analogous pathogens [18].

To curtail these risks, host sites have at their disposal an arsenal of public health measures that they can used independently or concurrently. Such measures can act at three different stages: at entry, on-site and at exit. Exit controls are an efficacious way to disrupt the global spread of infectious pathogens [58]. However, they are rarely explicitly used because the onus is then on the exit-screening country to treat the detected case. Regarding the world cup, visitors were returning to a large number of home locations, making the assessment of exit controls difficult. For these reasons exit controls were not included in our evaluation.

Instead, we focused our study on the role of the most common stages of control: entry and on-site controls. We used the example of the recent FIFA World Cup in Qatar to investigate the effect of different types of interventions, namely, vaccination, antigen and RT-PCR testing, with the testing taking place at different stages in a participant's travel to or sojourn in the host location. We made the realistic assumption that travellers are vaccinated prior to their arrival in the host location. We focused the implementation of interventions

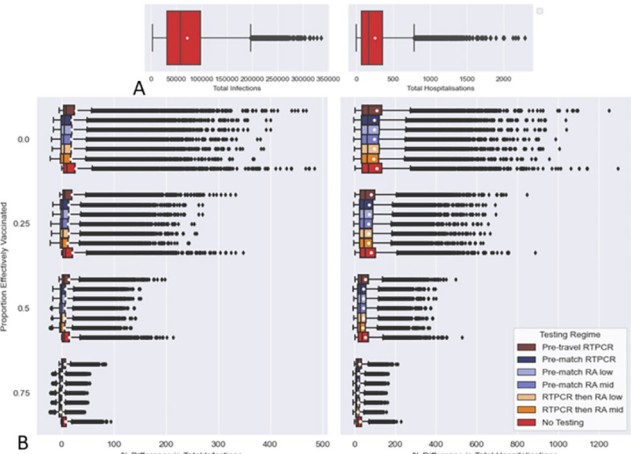

**Fig 5. Comparison of a policy ensuring all visitors must be effectively vaccinated but not having testing "effective visitor vaccination") against other policies.** A: Boxplots of Total Infections and Hospitalisation under "effective visitor vaccination" ($v_A = v_B = 1$). B Boxplots of % relative differences in Total Infections and Hospitalisation seen under various testing regimes at differing levels of effective vaccination for visitors compared to "effective visitor vaccination" as a control. In B % relative differences are calculated between simulations made with the same Latin Hypercbe (LH) sample, see Eq 4. Testing regimes used in comparisons are "No Testing", "Pre-Travel RT-PCR", "Pre-Match RT-PCR", "Pre-Match RA" or "RT-PCR then RA" testing regimes (see Table 7). Levels of effective vaccination for visitors in the comparisons are $v_A = v_B = 0$, $v_A = v_B = 0.25$, $v_A = v_B = 0.5$ and $v_A = v_B = 0.75$. The white dots on the boxplots represent mean values. All parameters other than those relating to effective vaccination for visitors ($v_A$ and $v_B$) are drawn using LH sampling from distributions outlined in Tables 2, 3 and 5.

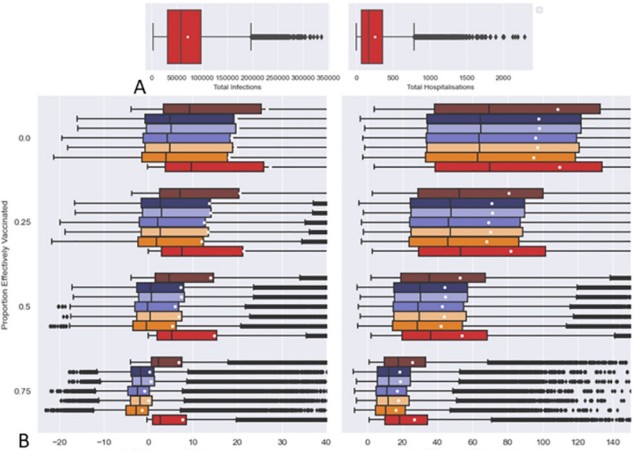

**Fig 6. Comparison of a policy ensuring all visitors must be effectively vaccinated but not having testing "effective visitor vaccination") against other policies.** A: Boxplots of Total Infections and Hospitalisation under "effective visitor vaccination" ($v_A = v_B = 1$). B Boxplots of % relative differences in Total Infections and Hospitalisation seen under various testing regimes at differing levels of effective vaccination for visitors compared to "effective visitor vaccination" as a control. Note that Figs 5B and 6B plot the same data, Fig 6B simply has a decreased range on the x-axis to aid comparison between boxplots. In B % relative differences are calculated between simulations made with the same Latin Hypercbe (LH) sample, see Eq 4. Testing regimes used in comparisons are "No Testing", "Pre-Travel RT-PCR", "Pre-Match RT-PCR", "Pre-Match RA" or "RT-PCR then RA" testing regimes (see Table 7). Levels of effective vaccination for visitors in the comparisons are $v_A = v_B = 0$, $v_A = v_B = 0.25$, $v_A = v_B = 0.5$ and $v_A = v_B = 0.75$. The white dots on the boxplots represent mean values. All parameters other than those relating to effective vaccination for visitors ($v_A$ and $v_B$) are drawn using LH sampling from distributions outlined in Tables 2, 3 and 5.

on spectators and those staffing the MGE rather than on the whole population of the host location.

We found that pre-travel testing in the FIFA world cup has little effect on disease burden, potentially due to the pre-existence of community transmission and leakage of COVID-19 false negative visitors from abroad. Indeed, when community transmission is already taking place, the contribution of introduced cases is minimal [59]. It can be inferred that in cases were disease is completely absent from the mass gathering site, pre-travel testing would prove beneficial, as has been observed in location which implemented a COVID-zero policy [60], but this was not evaluated here. We found that pre-match testing was more effective, with pre-match RT-PCR and pre match RA tests being comparable. We found only marginal improvements in COVID-19 control if visitors had undergone both pre-travel and pre-match testing.

We also investigated the relative roles of pre-match and pre-travel testing in comparison to requirements for visitors to be effectively vaccinated. We found that such a vaccination based policy generally outperformed pre-travel testing regimes in controlling infections. The "effective visitor vaccination" policy generally leads to similar reductions in infections compared to pre-match and "RT-PCR then RA" testing regimes, but is more likely to be an improvement in select circumstances. When it came to reducing hospitalisation such a policy more consistently outperformed testing regimes and often to a much greater extent. As the background levels of effective vaccination amongst visitors decreased, the reduction in hospitalisation under various testing regimes paled in comparison to reductions under a requirement that all visitors be effectively vaccinated.

The state of Qatar decided to remove COVID-19 pre-travel testing and vaccination related travel restrictions for the period of the World Cup. Instead merely suggesting that all visitors in this period be fully vaccinated and up to date on their booster doses [17, 50, 51]. Although, it should be mentioned that in order to access Qatari healthcare facilities visitors had to register their health status on the Ehetraz app [55]. Fig 7 demonstrates that the number of COVID-19 cases and hospitalisations had been on a downward trajectory before the World Cup. An increase in the number of COVID-19 cases and hospitalisations starts towards the end of the group stages, peaking at the beginning of the quarter final stage of the tournament. Such an increase may support [19], who found that there was little effect on COVID-19 transmission associated with a nation hosting a UEFA 2020 match, but speculated that hosting an entire tournament such as FIFA 2022 could increase COVID-19 transmission. The increase in cases and hospitalisations is then followed by a decline, most probably reflecting less interest from certain fan-bases as their national side drops out of the tournament. Our work here would suggest that the State of Qatar's removal of pre-travel testing may have been reasonable. However, the resulting spike in COVID-19 cases and hospitalisations may have been avoided with the enforcement of a policy requiring visitors to have had a second dose or a booster COVID-19 vaccination within a reasonable time-frame (e.g., 6 months to 14 days) prior to entry. Thus, ensuring COVID-19 vaccination among visitors was actually effective [26, 61–62].

Daily Qatari data on the number of COVID-19 detections, hospitalisations and vaccinations differentiating between second and third (booster dose) required to fit our model is limited. The data-set from the State of Qatar [64] is missing data between 27-10-2021 and 29-6-2022, the data is patchy for June through August 2022 and no record was made to indicate if a vaccine dose was a second or third booster. When it comes to Qatar 'Our World in Data' [48] only lists COVID-19 case detections, missing the data on hospitalisations and vaccinations that were also required to fit our model. Furthermore, we did not have access to estimates of staffing (stadium or policing) and numbers of spectators for matches (including their composition by nationality) from the State of Qatar. Therefore, we chose to use a scenario analyses based on LHS. If the required data was available, our scenario analyses could have been based

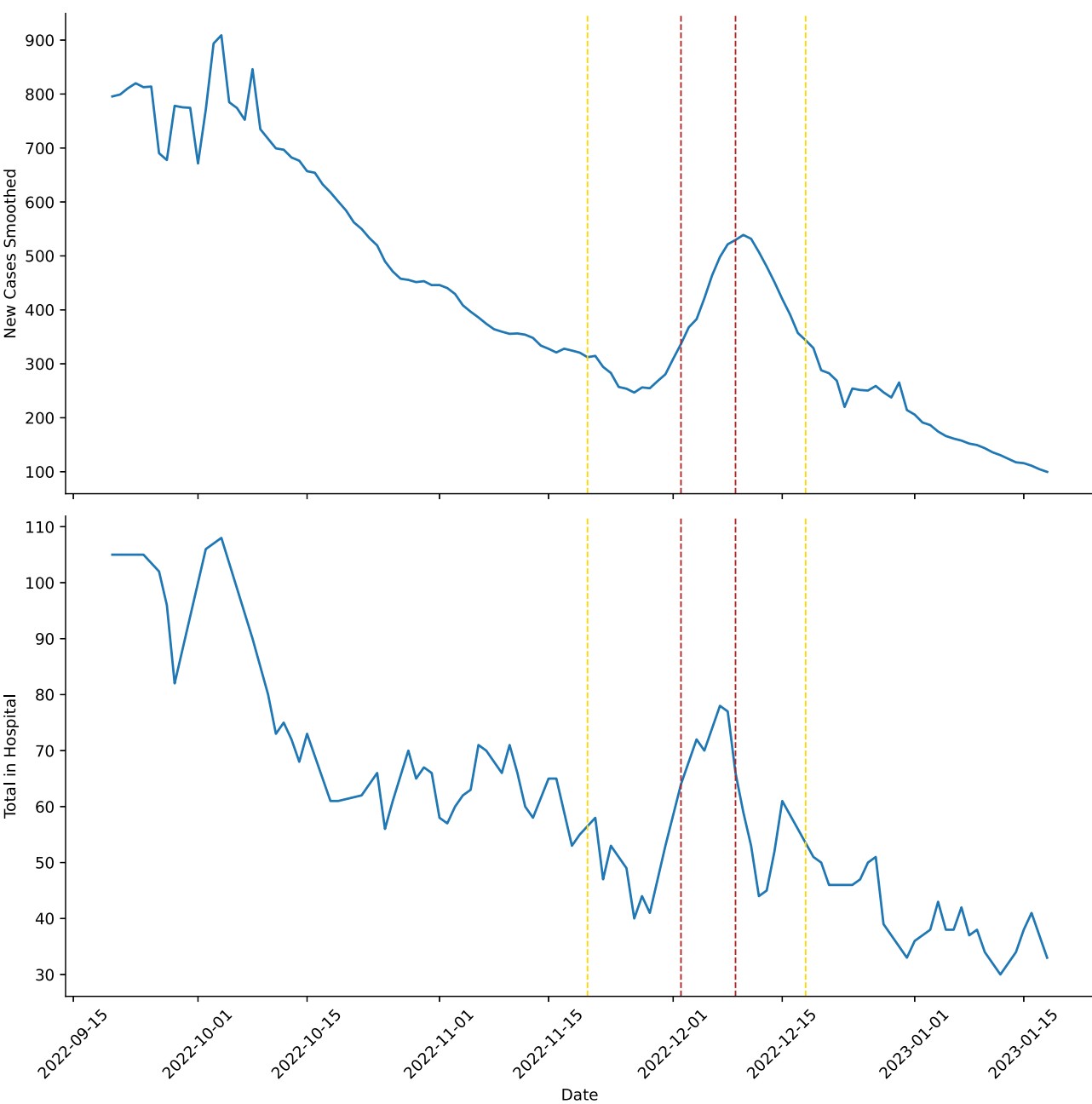

**Fig 7. Qatari COVID-19 New Cases Smoothed [48] and Acute Cases under Hospital Treatment [64] around the time of the World Cup.** The area between the yellow dotted lines is the time between the first world cup match and the final match. The area between the red dotted lines is the time between the last group stage match and the beginning of the quarter finals.

on a two stage approach. The first stage would have been to fit a single host cluster version of the model to Qatari COVID-19 detections, hospitalisations and vaccinations. The second stage would have been to use the parameters and variable estimates from the fitting in stochastic processes, such as $\tau$-leap methods [65], to simulate the scenarios. The large number of LHS samples used for our scenario analyses rendered the use of such stochastic simulations impractical, at least with numbers of stochastic simulations large enough to control for the resulting

aleatoric uncertainty. However, a fitting stage would have decreased the epistemic uncertainty, making such large numbers of stochastic simulations feasible.

A number of limitations of our work come from an attempt to reduce the parameter space being sampled, the number of scenarios being modelled and model complexity. To reduce the parameter space being sampled, we assumed isolation is as effective at reducing transmission from visitors as residents. Adding differential effects of isolation on transmission between hosts and visitors would have added more complexity to our model and increased the parameter space being sampled. A scenario with fewer clusters of people would present a more simplified and therefore ideal setting for assessing the effect differential group isolation on transmission, using our generalised model framework.

To reduce the number of scenarios being modelled, matches involving the host Qatari team were excluded from our modelling. Including such matches would require a second set of scenarios to be modelled. This approach would replace the two sets of visitor clusters with a single set of visitor clusters and corresponding increase in the population of the "Host Spectators" cluster. This set of scenarios would require simulations with another set LH sampled parameters, parameters surrounding a second set of visitors having being removed ($\sigma_B$ and $\nu_B$, see Table 5). Furthermore, ending transmission at day 7 of simulations would need to be reconsidered (see Table 6), in order to better approximate post match disease dynamics in the host nation. This may also require that the differential effectiveness of isolation between clusters be considered as well (see above).

To reduce model complexity, we excluded the possibilities of isolation upon development of symptoms or isolation on a positive test as a result of symptoms. The former possibility could come about through a policy of "Isolation upon Symptoms". The latter possibility through a "Test upon Symptoms" regime. Both would be worth comparing to the testing regimes and "Effective Vaccination" policy studied here. However, to model either possibility rates of flow from the pre-symptomatic state ($P_{Iiv}$) to either of the corresponding isolation clusters mid-stage symptomatic infection states, $M_{Iiv}$ or $M_{Hiv}$, would need to be added to the model. For RT-PCR tests being used in this circumstance the rates of flow would instead lead to the associated "Waiting for Positive RT-PCR" cluster and the effect of isolation of this cluster would need to be considered. The testing regimes studied here were discrete mandated events, this led us to assume full compliance. With "Isolation upon Symptoms" the new rates of flow from the pre-symptomatic state ($P_{Iiv}$) would be governed by an isolation compliance parameter. For a "Test upon Symptoms" regime this isolation compliance parameter would be multiplied by the test's sensitivity to form a super-parameter. In both cases the uncertainty of the isolation compliance parameter would have to be evaluated and a new model sensitivity analyses performed.

## 5 Conclusion

Our study demonstrates the feasibility of using modelling to assess disease control strategies at large MGEs, such as the FIFA World Cup 2022, in a time of COVID-19 and other pandemics. We find that requiring visitors to be effectively vaccinated is more effective than visitor pre-travel COVID-19 testing, and typically outperforms pre-event COVID-19 testing of attendees. Differing conclusions may be drawn if COVID-19 transmission was absent from the host nation [60]. Therefore, the State of Qatar's abandonment of pre-travel COVID-19 testing may have been reasonable. However, in light of the COVID-19 cases and hospitalisations seen over the world cup we conclude that pre-travel COVID-19 testing should have been replaced with required effective vaccination pre-entry. Put another way, all visitors should have completed a primary series of vaccination close to the time of entry or should have had a booster dose timed so as to ensure the fullest possible immunity.

## Supporting information

**S1 Methods. Deviation of the Basic Reproductive Number, $R_0$, and its relationship to the transmission term, $\beta$.**
(PDF)

**S1 Results. In depth examinations of % relative differences and PRCCs.**
(PDF)

**S1 Table. Differences in PCCs between proportions of Team A visitor effectively vaccinated and different testing regimes.** Differences are transformed to z-scores using methods outlined in [56].
(CSV)

**S2 Table. Differences in PCCs between proportions of Team B visitor effectively vaccinated and different testing regimes.** Differences are transformed to z-scores using methods outlined in [56].
(CSV)

## Acknowledgments

Thanks to Karishma Changlani for proof reading the revised version of this manuscript.

## Author Contributions

**Conceptualization:** Martin Grunnill, Julien Arino, Zachary McCarthy, Laurent Coudeville, Edward W. Thommes, Jianhong Wu.

**Data curation:** Martin Grunnill, Zachary McCarthy, Nicola Luigi Bragazzi, Amine Amiche.

**Formal analysis:** Martin Grunnill.

**Funding acquisition:** Julien Arino, Laurent Coudeville, Edward W. Thommes, Lydia Bourouiba, Jianhong Wu.

**Investigation:** Martin Grunnill.

**Methodology:** Martin Grunnill, Julien Arino, Zachary McCarthy, Nicola Luigi Bragazzi, Laurent Coudeville, Edward W. Thommes, Amine Amiche, Abbas Ghasemi, Mohammadali Tofighi, Ali Asgary, Jianhong Wu.

**Project administration:** Martin Grunnill, Zachary McCarthy, Jianhong Wu.

**Resources:** Martin Grunnill, Julien Arino, Jianhong Wu.

**Software:** Martin Grunnill.

**Supervision:** Jianhong Wu.

**Validation:** Martin Grunnill.

**Visualization:** Martin Grunnill, Laurent Coudeville, Lydia Bourouiba.

**Writing – original draft:** Martin Grunnill, Julien Arino, Nicola Luigi Bragazzi.

**Writing – review & editing:** Martin Grunnill, Julien Arino, Amine Amiche, Abbas Ghasemi, Lydia Bourouiba, Mortaza Baky-Haskuee, Jianhong Wu.

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
