## [Decision Letter · Decision Letter 0]

29 Jun 2023

Dear Dr Grunnill,

Thank you very much for submitting your manuscript "Modelling Disease Mitigation at Mass Gatherings: A Case Study of COVID-19 at the 2022 FIFA World Cup." for consideration at PLOS Computational Biology.

As with all papers reviewed by the journal, your manuscript was reviewed by members of the editorial board and by several independent reviewers. In light of the reviews (below this email), we would like to invite the resubmission of a significantly-revised version that takes into account the reviewers' comments.

The Authors are expected to address all the criticisms by all Reviewers. In particular, please provide further explanation on the model parameters (Reviewers #1 and #2), and reconsider if fixing the VE waning effect in the sensitivity analysis would lead to increasing protection over time (Reviewer #1). In additional to the above comments, please address,

1. Could the development of respiratory symptoms trigger testing in practice? Would that affect the study findings?

2. R0 was assumed to be 2 to 7, referring to a paper published in 2021. However, the Omicron variant was predominantly circulating in the study period, with an R0 of around 10. The assumed range was too low and should be reassessed.

We cannot make any decision about publication until we have seen the revised manuscript and your response to the reviewers' comments. Your revised manuscript is also likely to be sent to reviewers for further evaluation.

Sincerely,

Eric HY Lau, Ph.D.

Academic Editor

PLOS Computational Biology

Virginia Pitzer

Section Editor

PLOS Computational Biology

The Authors are expected to address all the criticisms by all Reviewers. In particular, please provide further explanation on the model parameters (Reviewers #1 and #2), and reconsider if fixing the VE waning effect in the sensitivity analysis would lead to increasing protection over time (Reviewer #1). In additional to the above comments, please address,

1. Could the development of respiratory symptoms trigger testing in practice? Would that affect the study findings?

2. R0 was assumed to be 2 to 7, referring to a paper published in 2021. However, the Omicron variant was predominantly circulating in the study period, with an R0 of around 10. The assumed range was too low and should be reassessed.

Reviewer's Responses to Questions

**Comments to the Authors:**

Reviewer #1: In this manuscript the authors extend their existing model (used to simulate meningococcal infections during Hajj) into a general framework for simulating infections at mass-gathering events, and use this framework to estimate how effective various testing strategies and vaccination requirements might have been at reducing COVID-19 infections and hospitalisations during the 2022 FIFA World Cup (held in Qatar).

The manuscript is generally clear, the framework is described in detail, the analyses incorporate many sources of uncertainty, and the findings are sensible.

I have several comments, primarily regarding details of the modelling framework and the reported analyses.

1. In section 2.2 the authors state:

> We chose to model possible matches from the FIFA 2022 World Cup (not involving the Qatari team). Each match is seen as a 7 day MGE.

It took me quite a while to realise that this meant that each match was treated as an *entirely independent* MGE, rather than considering the World Cup as a single timeline over which many MGEs occurred. On my initial read through the paper and supplementary materials, I was searching for the match schedule in order to understand the potential interactions between matches. In retrospect, this detail is implicit in the above remarks and in Table 6, but I feel it would be helpful to emphasise this and help future readers avoid my confusion.

2. Were matches involving the Qatari team excluded as a consequence of assuming that the populations of nations A and B are distinct from the local Qatari population? If so, it seems like an unfortunate limitation of the framework. In the discussion the authors state:

> Adding differential effects of isolation on transmission between hosts and visitors would have added more complexity to our model and increased the parameter space being sampled.

This is entirely fair, but would it be possible to extend the framework in the future to allow for matches involving the host population without increasing the parameter space? If so, this would be worth identifying in the discussion section.

3. In section 2.2 the authors state:

> The eight stadiums hosting matches have estimated capacities ranging from 40,000 to 80,000 (45). We assume therefore that the population attending simulated fixtures ranges from 4,000 to 80,000.

Were there World Cup matches where attendance was as low as 10% of the stadium capacity? Or was this lower bound simply a means of exploring parameter uncertainty?

4. In section 2.2.2 the authors state:

> LH sampling was done using uniform distributions and a sample size of 10,000.

It might be helpful to mention that parameters were sampled from uniform distributions in the captions for Tables 2, 3, and 5. I didn't initially appreciate that this was true for all of the listed parameters. In particular, for the infection prevalence and effectively-vaccinated parameters for nations A and B, I initially thought that the values may have been informed by nation-specific estimates for each match.

5. In Table 3, the vaccine effectiveness against infection for the *waned* vaccination group is fixed at 0.2230. This value is *higher* than the lower bound for vaccine effectiveness against infection for those *effectively* vaccinated (0.1730). So in about 8% of the simulations, vaccine-acquired immunity would seem to increase over time. Is that correct?

6. In Figure 3B, for a small proportion of simulations the testing regimes resulted in greater numbers of infections and hospitalisations than in simulations with no testing regime — the percentage difference isn't always negative. Since the model is deterministic, and the same Latin Hypercube samples were used for each pair of no-testing/with-testing simulations (right?), I'm struggling to understand how this might occur. Is it possible that the initial state differed between the no-testing and with-testing simulations? Or have I missed an obvious explanation?

7. In the discussion, with reference to Figure 6, the authors state:

> The data-set from the (63) [acute cases under hospital treatment] is missing data between 27-10-2021 and 29-6-2022, the data is patchy after 29-6-2022 and no record was made to indicate if a vaccine dose was a second or third booster.

The missing data are not evident in Figure 6, the "Total in Hospital" time-series has no apparent gaps or discontinuities. Were the missing data limited to specific details about each case during this time period, rather than absence of cases?

8. The authors have made all of their code available in a public GitHub repository, which is great to see. I have explored it a little (e.g., to confirm that all parameter values were being sampled from uniform distributions). The code is generally clear and well structured, for which I thank them!

I noticed that a number of data files are read using hard-coded paths that are specific to an author's computer (e.g., `C:/Users/mdgru/[...]`) rather than using paths that are relative to the repository (e.g., `../parameters/data_extraction/[...]`). This means that some of the code will likely fail to run on other people's computers without some small modifications. I'm pointing this out because writing reproducible code can be surprisingly hard and we rarely receive feedback.

Also, it would be great if the authors could add a license to the repository (see, e.g., https://choosealicense.com/) so that other people are able to use the code without any potential for copyright issues.

9. Are the authors planning to release a generic implementation of this mass-gathering event framework, in addition to the 2022 World Cup analyses that have been provided in the public GitHub repository?

Minor comments:

1. In Table 2, the values for "Efficacy of vaccination with regards to hospitalisation for those effectively vaccinated" could be described as "0.837 to 1", to be consistent with parameters and avoid potential confusion about the dash being a minus sign.

2. In Table 3, I understand the rationale for most of the vaccine rate parameters being set to zero, but shouldn't the "Rate of waning immunity of vaccination" be non-zero?

3. The abstract and author summary both end with the following remark:

"[...] a policy requiring visitors to have had a recent COVID-19 vaccination may have prevented the increase in COVID-19 cases and hospitalisations during the world cup."

Comparing Figure 3A ("Total Infections and Hospitalisation in simulations made with no testing regime") and Figure 5A ("Boxplots of Total Infections and Hospitalisation under 'effective visitor vaccination'") suggests that such a policy would have reduced cases and hospitalisations. It's less clear to me that this policy would have entirely prevented an increase in cases and hospitalisations during the world cup.

For reference, the (smoothed) cases and hospitalisations peaked at around 600 and 80, respectively (shown in Figure 6). It isn't possible to directly compare these peak values to the total infections and hospitalisations shown in Figures 3A and 5A, since those totals are calculated over a 100-day window.

Reviewer #2: The authors analysed the various COVID-19 mitigation strategies in the context of mass gathering events at the FIFA World Cup. This involved substantial extensions of an existing ODE model and a simulation-based study with Latin Hypercube Sampling over some parameters. The model is sufficiently detailed and well explained. The FIFA World Cup scenario was an appropriate setting for the use of this kind of metapopulation model. The paper demonstrates the ability of a simulation study to influence policy, though, admittedly the authors did not have access to data for model fitting. I consider this work publishable, however, the choice of which parameters were reported with uncertainty seemed arbitrary so the paper could benefit from a more thorough sensitivity analysis. In particular, changing the test sensitivities would likely give rather different results (noting the RA sensitivity comes from a study which reports that 0.728 is likely an overestimate).

Minor comments:

1. Equation (1) does not include waning immunity (i.e. the equations are invalid for v=2).

2. Tables 4 and 7 are difficult to read (alternate shading of rows could help here).

3. pg. 10 has a typographical error in the bounds for N_A.

4. I’m not sure why Figure 5 has a ‘B’ and ‘C’ section, I think they could be combined.

**Have the authors made all data and (if applicable) computational code underlying the findings in their manuscript fully available?**

Reviewer #1: Yes

Reviewer #2: Yes

PLOS authors have the option to publish the peer review history of their article (what does this mean?). If published, this will include your full peer review and any attached files.

Reviewer #1: **Yes: **Robert Moss

Reviewer #2: No
---

## [Decision Letter · Decision Letter 1]

17 Oct 2023

Dear Dr Grunnill,

Thank you very much for submitting your manuscript "Modelling Disease Mitigation at Mass Gatherings: A Case Study of COVID-19 at the 2022 FIFA World Cup." for consideration at PLOS Computational Biology. As with all papers reviewed by the journal, your manuscript was reviewed by members of the editorial board and by several independent reviewers. The reviewers appreciated the attention to an important topic. Based on the reviews, we are likely to accept this manuscript for publication, providing that you modify the manuscript according to the review recommendations.

The Authors have addressed most of the comments. There are two minor issues,

1. Would the authors consider applying for open source license on your code to promote sharing and usability?

2. Figure 5B, the authors may consider using the same axis range or other presentation to facilitate comparison between % vaccinated.

Sincerely,

Eric HY Lau, Ph.D.

Academic Editor

PLOS Computational Biology

Virginia Pitzer

Section Editor

PLOS Computational Biology

The Authors have addressed most of the comments. There are two minor issues,

1. Would the authors consider applying for open source license on your code to promote sharing and usability?

2. Figure 5B, the authors may consider using the same axis range or other presentation to facilitate comparison between % vaccinated.

Reviewer's Responses to Questions

**Comments to the Authors:**

Reviewer #1: I thank the authors for thoroughly addressing my comments, both in their response letter and in the revised manuscript, and for resolving several points of confusion on my part. I have only a few minor comments regarding this revised manuscript.

1. Thank you for revising the uncertainty and sensitivity analyses to prevent vaccine-acquired immunity from growing stronger over time, and for using identical initial states in the no-testing and with-testing simulations.

The revised version of Figure 3 looks great.

2. Thank you for explaining why the vaccine waning rate in Table 3 is set to zero, and remarking on this in the text (section 2.2).

It might be a useful reminder to also add this remark to the "Sources" column in Table 3, in case the reader fails to notice the revised sentence.

3. Thank you for revising the code in the public repository and adding some very helpful instructions to `README.md`, which is fantastic.

Regarding the choice of a license for the code, the authors responded:

> We are looking into this.

The code is available in a public GitHub repository, so under the GitHub Terms of Service, other GitHub users are allowed to view and fork the repository.

But without an explicit license, no one may reproduce, distribute, or create derivative works from your code.

Please note that adding a license allows others to reuse your code, but you retain the copyright.

**Have the authors made all data and (if applicable) computational code underlying the findings in their manuscript fully available?**

Reviewer #1: Yes

PLOS authors have the option to publish the peer review history of their article (what does this mean?). If published, this will include your full peer review and any attached files.

Reviewer #1: **Yes: **Rob Moss

Figure Files:

Data Requirements:

Reproducibility:

References:

---

## [Editor Report · Decision Letter 2]

20 Nov 2023

Dear Dr Grunnill,

We are pleased to inform you that your manuscript 'Modelling Disease Mitigation at Mass Gatherings: A Case Study of COVID-19 at the 2022 FIFA World Cup.' has been provisionally accepted for publication in PLOS Computational Biology.

Best regards,

Eric HY Lau, Ph.D.

Academic Editor

PLOS Computational Biology

Virginia Pitzer

Section Editor

PLOS Computational Biology

Thanks for addressing all the editor’s and reviewers' comments. Congratulations on the excellent work!

---

## [Editor Report · Acceptance letter]

15 Dec 2023

PCOMPBIOL-D-23-00394R2 

Modelling Disease Mitigation at Mass Gatherings: A Case Study of COVID-19 at the 2022 FIFA World Cup.

Dear Dr Grunnill,

I am pleased to inform you that your manuscript has been formally accepted for publication in PLOS Computational Biology. Your manuscript is now with our production department and you will be notified of the publication date in due course.

With kind regards,

Bernadett Koltai
